# Do Neighborhood Factors Modify the Effects of Lead Exposure on Child Behavior?

**DOI:** 10.3390/toxics10090517

**Published:** 2022-08-31

**Authors:** Seth Frndak, Gabriel Barg, Elena I. Queirolo, Nelly Mañay, Craig Colder, Guan Yu, Zia Ahmed, Katarzyna Kordas

**Affiliations:** 1Department of Epidemiology and Environmental Health, University at Buffalo—State University of New York, New York, NY 14203, USA; 2Department of Neuroscience and Learning, Catholic University of Uruguay, Montevideo 11600, Uruguay; 3Faculty of Chemistry, University of the Republic of Uruguay (UDELAR), Montevideo 11600, Uruguay; 4Department of Psychology, University at Buffalo—State University of New York, New York, NY 14214, USA; 5Department of Biostatistics, University of Pittsburgh, Pittsburgh, PA 15260, USA; 6Research and Education in Energy, Environment and Water (RENEW) Institute, University at Buffalo—State University of New York, New York, NY 14260, USA

**Keywords:** neighborhood disadvantage, greenspace, effect modification, blood lead levels, child behavior

## Abstract

Lead exposure and neighborhoods can affect children’s behavior, but it is unclear if neighborhood characteristics modify the effects of lead on behavior. Understanding these modifications has important intervention implications. Blood lead levels (BLLs) in children (~7 years) from Montevideo, Uruguay, were categorized at 2 µg/dL. Teachers completed two behavior rating scales (*n* = 455). At one-year follow-up (*n* = 380), caregivers reported child tantrums and parenting conflicts. Multilevel generalized linear models tested associations between BLLs and behavior, with neighborhood disadvantage, normalized difference vegetation index (NDVI), and distance to nearest greenspace as effect modifiers. No effect modification was noted for neighborhood disadvantage or NDVI. Children living nearest to greenspace with BLLs < 2 µg/dL were lower on behavior problem scales compared to children with BLLs ≥ 2 µg/dL. When furthest from greenspace, children were similar on behavior problems regardless of BLL. The probability of daily tantrums and conflicts was ~20% among children with BLLs < 2 µg/dL compared to ~45% among children with BLLs ≥ 2 µg/dL when closest to greenspace. Furthest from greenspace, BLLs were not associated with tantrums and conflicts. Effect modification of BLL on child behavior by distance to greenspace suggests that interventions should consider both greenspace access and lead exposure prevention.

## 1. Introduction

Blood lead levels (BLLs) in children have declined worldwide. For example, recent estimates of mean BLLs among 1–11 year-old children from the United States have fallen to <1 μg/dL [1]. In response to persistent negative associations between low-level BLLs and child development, the Centers for Disease Control and Prevention lowered the actionable BLL from 5 to 3.5 μg/dL in 2021 [2]. Because the effects of lead on child development are not homogenous [3], it is important to clarify for whom lead exposure has the most significant adverse effects.

A complex interrelationship exists between childhood lead exposure, the neighborhood environment, and child behavior [4,5]. Lead exposure is associated with externalizing and internalizing behaviors, and poor executive functions even at BLLs as low as 5 µg/dL. Lead affects behavioral development in children by hindering neurologic synaptogenesis, disrupting neurotransmitters and increasing oxidative stress [6]. Among children aged ~7 years from Montevideo, Uruguay, low BLL exposure (70% < 5 µg/dL) was associated with poorer executive functioning overall [7] and greater problems of inattention among children with iron deficiency [8]. Other studies have further confirmed the association between BLLs and child behavior [4,9,10,11,12,13]. Neighborhood factors also influence child behavior. Access to greenspace, parks [14,15,16] and social capital [17,18] are associated with lower child behavior problems. Conversely, exposure to neighborhood violence [19,20] and neighborhood disadvantage [21,22,23] are associated with greater behavior problems. There is an overlap among neighborhood factors associated with behavior problems and those related to lead exposure, including racial composition [24,25], crime rates [26], and neighborhood poverty [21]. While many studies treat neighborhood factors as confounders of the lead–behavior relationship, neighborhood factors could be examined, instead, as effect modifiers [27].

Heterogeneity in neurotoxic effects of lead on child behavior may be due to differences in neighborhood sources of stress or enrichment [3,27]. Children from disadvantaged neighborhoods experience higher levels of stress, more stressful life events [28], and subsequent hypothalamic adrenal (HPA) axis dysregulation [29], potentially exacerbating the neurotoxic effects of lead. A combination of lead exposure and low socioeconomic position is associated with low performance on cognitive and developmental assessments among children [30,31,32,33]. Animal studies also support this hypothesis [34,35]. Conversely, enriching neighborhood factors may buffer adverse effects of lead. Neighborhood greenspace and park access are associated with positive childhood adjustment [14,36,37,38]. In animal studies, rats exposed to lead performed better on learning tasks if they were raised in an enriched environment compared to rats raised in isolation [39,40,41].

Leveraging data from the Salud Ambiental Montevideo (SAM) cohort of schoolchildren from Montevideo, Uruguay, we investigated the association between BLLs measured at ~7 years of age and (1) teacher ratings of child behavior collected at the same time (cross-sectional study), and (2) caregiver reports of behavior problems, including temper tantrums and conflicts with parents, completed at ~8 years of age (longitudinal study). Based on addresses collected at ~7 years, we created measures of neighborhood disadvantage, distance to nearest greenspace and average normalized difference vegetation index (NDVI). We hypothesized that children with BLLs ≥ 2 µg/dL would demonstrate greater mean behavior problem scores (worse behavior) in neighborhoods with greater disadvantage compared to children with BLLs < 2 µg/dL in similar neighborhoods. Furthermore, children with BLLs ≥ 2 µg/dL would have similar mean behavior scores to children with BLLs < 2 µg/dL in neighborhoods with high NDVI and greenspace.

## 2. Materials and Methods

### 2.1. Sample Recruitment and Analytical Samples

Recruitment for SAM occurred between 2009 and 2019 in Montevideo, Uruguay. From 2009–2013, we contacted private elementary schools in areas of suspected heavy metal exposure, with a special focus on lead [42,43]. Between 2015–2019, the Uruguayan Ministry of Education granted permission to recruit public-school children. Subsequently, we expanded recruitment via posters hung in public areas. We focused on schools classified from 3–5 based on Uruguayan socioeconomic school classification: 1 (highest) to 5 (lowest). Between 2009–2013, all evaluations were performed at the child’s school. From 2015–2019, evaluations were completed at the Catholic University of Uruguay. Caregivers provided consent directly after an information session or later after deliberation with family members. All caregivers provided written consent, and children gave verbal assent. Institutional review boards of the University at Buffalo and the Catholic University of Uruguay approved all study protocols. By 2019, 856 non-sibling, 1st grade children (~7 years of age) from 55 schools and 220 census segments were enrolled into the SAM study.

Children were included in the analytical sample for the cross-sectional study if they had complete location data, blood lead measurements, and complete teacher reports of problem behaviors. A total of 97 (11%) children did not have BLLs, and an additional 17 (2%) children did not have location data. Because laboratory methods for detecting low levels of lead improved during data collection, we removed participants with a limit of detection (LOD) >2 µg/dL (*n* = 57). Our final eligible sample was 685 participants. Within the eligible sample, 230 (34% of eligible) did not have complete teacher-reported behavior, resulting in a final analytical cross-sectional sample of 455 participants. Due to potential selection effects, differences between the analytical sample (*n* = 455) and the non-selected participants (*n* = 230) were tested as described below.

Non-sibling participants who were enrolled in SAM between 2015 and 2019 were eligible for annual follow-up (*n* = 512). Among those who completed the first annual follow-up (~8 years of age), 63 (12%) did not have wave 1 BLL data, 14 (3%) did not have location data, and 3 (1%) had limit of detection above 2 µg/dL. Our final eligible sample was 432. Within the eligible sample, 52 (12%) participants did not complete the caregiver behavior report or were lost to follow-up. The final analytical sample for the longitudinal study was 380 participants. The differences between this analytical sample (*n* = 380) and the non-selected participants (*n* = 52) were tested for selection effects as outlined below.

### 2.2. Measures

#### 2.2.1. Behavior Problem Measures in the Cross-Sectional Study

The child’s teacher completed two behavior scales: the Conners’ Teachers Rating Scale—Revised, Short Form (CTRS-R:S) [43,44] and the Behavior Rating Inventory of Executive Function (BRIEF) [45]. For the CTRS-R:S, teachers rated child behavior over the previous month with 28 items across four scales: oppositional problems, hyperactivity, cognitive problems, and the attention deficit hyperactivity disorder (ADHD) index. All items were rated by the teacher according to frequency: never or rarely, occasionally, often, and frequently. Internal consistency (Cronbach’s alpha) of all CTRS-R:S scales was excellent in the analytical cross-sectional sample (oppositional problems = 0.90; hyperactivity = 0.95; cognitive problems = 0.93; ADHD index = 0.95). The original BRIEF scale contains 86 items. Based on feedback from the teachers regarding excess burden completing the surveys, only two scales were administered after 2014: inhibitory problems and planning/organizing problems. These two scales also had excellent internal consistency in our sample (inhibitory problems = 0.97; planning and organization problems = 0.95). Further details on the administration of the CTRS-R:S and BRIEF scales and their content are provided in our previous publication [6]. We used the CTRS-R:S and BRIEF age and sex normed T-scores.

#### 2.2.2. Behavior Measures in the Longitudinal Study

At one-year follow-up (child age ~8 years), caregivers provided answers to two questions regarding the child’s behavior. The first question was: “In the last 3 months, how often has your child had a tantrum?”. Response options were “more than once per day”, “almost every day”, “at least once per week”, “less than once per week”, or “never”. Reponses were dichotomized as “almost every day” or more vs. “at least once per week” or less. The second question was: “In the last 3 months, how often do you have a ‘battle of wills’ with your child?” Response options were: “never”, “rarely”, “sometimes” or “frequently”. Responses were dichotomized as having or not having frequent conflicts (“battle of wills”).

#### 2.2.3. Clinical Measures: BLL, Hemoglobin and Body Mass Index (BMI)

Blood samples were collected between 8 and 11 a.m. by a nurse phlebotomist at the participant’s school (years 2009–2013) or at the Catholic University of Uruguay (2015–2019) after a morning fast. Specific methodology of blood draw and laboratory analysis, including quality control measures, is provided in previous publications [6,46,47]. Child BLL was categorized as <2 µg/dL and ≥2 µg/dL. Different methods of atomic absorption spectrometry (AAS) were used throughout data collection. For our final analytical sample, flame ionization (VARIAN SpectrAA-55B) was used in 29 (6%) blood samples (LOD 1.8 µg/dL), graphite furnace (Thermo ICC 3400) in 237 (52%) samples (LOD 1.0 µg/dL from 2013–2016 and 0.4 µg/dL from 2016–2019) and graphite furnace (VARIAN SpectrAA-55B) in 189 (42%) samples (LOD 2.0 µg/dL from 2009–2010 and 0.80 µg/dL from 2010–2016). All samples in the longitudinal study 432 (100%) were analyzed via graphite furnace (Thermo ICC 3400 LOD 1.0 µg/dL from 2015–2016 and LOD 0.4 µg/dL from 2016–2019).

Hemoglobin was measured during the blood draw with a portable hemoglobinometer and expressed in g/dL. As outlined by the manufacturer (HemoCue, Lake Forest, CA, USA), quality control checks were performed prior to sample testing with three levels of controls. BMI (kg/m^2^) was calculated using the average height and weight of each child, taken in triplicate by a study nurse.

#### 2.2.4. Caregiver Questionnaire

Caregivers were asked to complete a questionnaire on the child’s birthdate, child’s sex, maternal education, maternal employment status, caregiver smoking habits, and household assets and income. Age in months was calculated based on date of birth and enrollment date. Maternal education in years was derived from the highest level of education completed. Caregiver smoking status was coded as “yes” if either caregiver reported currently smoking and “no” if neither smoked. Factor analysis of yes/no answers to 15 household assets resulted in the summation of 5 household assets with the highest eigenvalue (DVD player, computer, car, washing machine, and landline phone) into a possessions score, ranging 0–5.

#### 2.2.5. HOME Inventory Score

During a scheduled visit to the participant’s home, a trained social worker administered the Home Observation for the Measurement of the Environment Inventory (HOME) [48]. The HOME inventory score is a 59-item measure that includes 8 scales with higher scores reflecting greater household enrichment. We used the global measure: HOME inventory score. Internal consistency of the HOME inventory score is 0.90 for 6–10 year old children [49].

#### 2.2.6. Neighborhood Measures

As described previously [46], we created a neighborhood disadvantage index via factor analysis of the demographic characteristics of all census segments within the city of Montevideo. This neighborhood disadvantage factor had good construct validity, being associated with maternal education, maternal age, HOME inventory score, and number of possessions of wealth. Neighborhood disadvantage was not associated with child BLLs. Our neighborhood disadvantage factor was also associated with greater mean oppositional behavior scores from the CTRS-R:S (administered in this study sample), and problem shifting and emotional control from the BRIEF (not administered in this study sample) [46]. Neighborhood disadvantage level was assigned to all participants based on the census segment where they resided at the time of study enrollment. The census segment population total was also assigned to each participant based on location of their household.

Normalized difference vegetation index (NDVI) is a measure of green foliage derived from satellite imagery [50]. NDVI is calculated as: $\frac{NIR\;-\;RED}{NIR\;+\;RED}$, where *NIR* is near infra-red reflected light and *RED* is red reflected light. We used cloud-free images from the Planet Image archive [51]. Six summertime (Southern Hemisphere) dates were used for NDVI calculation: 24 December 2018; 28 January 2019; 19 February 2019; 23 December 2019; 28 January 2020; 23 February 2020. We used dates closer to the end of study recruitment because high-resolution (3 m rasters) imagery was only available after 2017. Change in NDVI in an urban setting is often measured in decades, not years, [52] lessening potential influence of measurement error. After obtaining NDVI scores for each raster, a 150 m buffer was created around the participant’s home. An average NDVI score for all rasters within this buffer zone, across all dates was created for our NDVI measure.

Greenspace was defined by the Intendencia de Montevideo Servicio de Geomática (IMSG) using aerial photogrammetry [53]. Greenspaces included parks, gardens, or landscaped areas. Distance to the nearest greenspace polygon was calculated in kilometers for each participant’s home address using the gDistance function in the rgeos R package [54].

### 2.3. Statistical Analysis: Cross-Sectional Study

#### 2.3.1. Selection Effects

Among the 685 participants eligible for the cross-sectional sample, only 455 participants had either BRIEF or CTRS-R:S data. These 455 participants are further divided into 362 participants with complete CTRS-R:S data and 448 participants with complete BRIEF data. To check for selection effects, differences were tested between those with (*n* = 362) and without (*n* = 323) CTRS-R:S data and those with (*n* = 448) and without (*n* = 237) BRIEF data. Differences were assessed using pairwise comparisons of complete case data, without imputation. All covariates, main exposure (BLL ≥ 2 µg/dL), and effect modifiers were tested for differences using either t tests for continuous data or chi-square tests for categorical data. To assess the level of difference and the potential impact of selection effects, Cohen’s d and *phi*-coefficients are provided for continuous and categorical data, respectively. Cohen’s d is interpreted as trivial (0.0–0.2), small (0.2–0.5) and moderate (0.50–0.80) and large (>0.80). *Phi*-coefficients can be either positive or negative in direction and are interpreted in the same way as a correlation coefficient: trivial (±0.0–0.2), small (±0.2–0.5), moderate (±0.5–0.7) and large (>±0.7).

#### 2.3.2. Missing Data

Imputation of missing covariate data in the cross-sectional study (*n* = 455 CTRS-R:S and BRIEF samples together) was performed using the missForest package in R [55]. Random forest imputation outperforms other imputation techniques and does not require aggregation across multiple imputed datasets [56]. Random forest is an extension of regression trees that utilizes multiple decision trees to form a final prediction. Because data were only missing at level-1, not level-2 (neighborhood-level), we did not consider multilevel imputation. Based on an iterative process testing the number of ntree (100–1000 stepping up by 100) and mtry (2–50) across random seeds, we set the number of ntree to 300 and mtry to 12. Our final random forest imputation model had a normalized root-mean-square error (NRMSE) of 0.002% (imputation error among continuous variables) and a percent falsely classified (PFC) error of 22.7% (imputation error among categorical variables). Both were within an acceptable range based on previous publications using this method [57,58,59,60].

#### 2.3.3. Multilevel Analysis

Separate cross-sectional analyses were performed for those with complete CTRS-R:S data (*n* = 362) and those with complete BRIEF data (*n* = 448). To account for slight skewness in the behavior T-scores (skewness range: 0.61–1.80) and clustered nature of the data (children nested within neighborhoods), we used multilevel models with a gamma distribution and an identity link using the glmer function in the lme4 R package [61]. To test variation among neighborhoods in child behavior, we reported intra-class correlation coefficients (ICCs) for each outcome. ICCs are calculated as the variation between neighborhoods divided by total variation as follows: BRIEF inhibitory control 0.18, BRIEF planning/organizing 0.13, CTRS-R:S oppositional behavior 0.23, CTRS-R:S cognitive problems 0.15, CTRS-R:S hyperactivity 0.26, CTRS-R:S ADHD index 0.25. Our choice of multilevel models was warranted given the moderate (~20%) variation in behavior due to neighborhood differences. To help with model estimation and convergence, the NDVI score was multiplied by 100 and census segment population divided by 1000. Because of the skewness of the distance to nearest greenspace (skewness = 2.98 in cross-sectional sample), we created deciles of this variable.

First, multilevel direct effects models tested for associations between BLL (≥2 and <2 μg/dL) and one of the three neighborhood factors. In these models, neighborhood disadvantage, NDVI, and distance to nearest greenspace (in deciles) were modeled as continuous variables. Each behavior scale T-score was modeled as a separate outcome. Therefore, 18 multilevel direct effects models were estimated. Confounders included: sex, age in months, year of enrollment, BMI, hemoglobin g/dL, caregiver smoking (yes/no), mother’s employment status (yes/no), HOME inventory score, maternal education in years, and number of possessions at the individual level (Level-1). Census segment population total was also included at the neighborhood level (Level-2). Next, we tested effect modification using an interaction term between BLL category and each neighborhood factor (neighborhood disadvantage, NDVI and distance to nearest greenspace were all modeled continuously). An additional 18 models with interaction terms were estimated.

### 2.4. Statistical Analysis: Longitudinal Study

We followed the procedure outlined above to assess the presence of selection effects in the longitudinal study, comparing those included in the study (*n* = 380) with the 52 participants without behavior data or lost to follow-up. Missing covariate data were also imputed using random forest imputation. In this case, we set ntree to 600 and mtry to 2. The resulting NRMSE was 0.001% and PFC was 5.7%. We also assessed the degree of between-neighborhood variation using an ICC calculation for discrete outcomes in multilevel modeling [62]: ICC = between group variance/(between group variance + π2/3). ICC for tantrums almost every day was 3% and for frequent conflicts, 2%. Based on these ICCs, we omitted multilevel modeling and used a generalized linear model with a binomial distribution and a logit link. Direct effects were modeled first for each outcome and neighborhood factor (six models). Interaction terms were then added separately (six additional models).

## 3. Results

### 3.1. Cross-Sectional Sample

Characteristics of the cross-sectional samples (BRIEF *n* = 448; CTRS-R:S *n* = 362) compared to non-selected participants (*n* = 237; *n* = 323, respectively) are presented in Table 1. Overall, our analytical samples were balanced between males and females (~55% male) aged ~7 years (82 months). Participants were of slightly lower socioeconomic status, having ~3 possessions and ~8.5 years of maternal education on average. At least one caregiver was a smoker in ~60% of the households, and most mothers were employed (~62%). Children had an average BMI of ~17.0 and average hemoglobin of ~13.0 g/dL. The prevalence of BLL ≥2 µg/dL was 52% in the BRIEF sample and 60% in the CTRS-R:S sample. The prevalence of clinical levels of behavior problems in this sample (T-scores ≥70) ranged from 10% to 23% across behavior scales.

Participants were from areas of higher neighborhood disadvantage (1.6 average standardized score) compared with an average Montevideo city census segment neighborhood disadvantage score of 0.0. Average NDVI was 0.34, and average distance to the nearest greenspace was 0.2 km. Behavior problems were relatively low, with mean T-scores for BRIEF and CTRS-R:S scales at ~50.0. Only BRIEF inhibitory control scores were slightly elevated (mean = 57, standard deviation = 14.9).

Differences were noted on the BRIEF/CTRS-R:S between analytical samples and the corresponding non-selected participants. First, children were ~1.5 months younger in the BRIEF/CTRS-R:S sample compared to non-selected participants (BRIEF t_682_ = 3.15, *p* < 0.01; CTRS-R:S t_682_ = 3.77, *p* < 0.01). These were small differences (BRIEF Cohen’s d = 0.25; CTRS-R:S Cohen’s d = 0.28). The BRIEF/CTRS-R:S sample had ~0.2 more possessions of wealth than the non-selected sample (BRIEF t_648_ = −2.33, *p* < 0.05; CTRS-R:S t_648_ = −2.01, *p* < 0.05), again these were small differences (BRIEF Cohen’s d = 0.15; CTRS-R:S Cohen’s d = 0.23). HOME inventory score was higher among the BRIEF/CTRS-R:S analytical samples compared to non-selected participants (BRIEF t_578_ = −3.23, *p* < 0.01; CTRS-R:S t_578_ = −3.38, *p* < 0.001), but these differences were small (BRIEF Cohen’s d = 0.27; CTRS-R:S Cohen’s d = 0.28). More children in the CTRS-R:S sample had BLL ≥2 µg/dL than among non-selected participants (CTRS-R:S χ_21_ = 35.37, *p* < 0.001), with trivial effect size (*phi* = 0.07). Children in the CTRS-R:S sample came from more populous census segments compared to non-selected participants, but the effect size was also trivial (t_683_ = −2.05, *p* = 0.04, Cohen’s d = 0.16).

Differences were observed between selected and non-selected children on neighborhood effect modifiers. Neighborhood disadvantage was greater in the BRIEF/CTRS-R:S analytical samples with a moderate effect size (BRIEF t_683_ = −6.24, Cohen’s d = 0.52, *p* < 0.001; CTRS-R:S t_683_ = −4.18, Cohen’s d = 0.32, *p* < 0.001). NDVI score was higher in the CTRS-R:S selected sample with moderate effect size (CTRS-R:S t_760_ = −2.80, Cohen’s d = 0.20, *p* = 0.01). Distance to the nearest greenspace was lower in the selected samples, with small effect size (BRIEF t_683_ = 4.36, Cohen’s d = 0.34, *p* < 0.001; CTRS-R:S t_683_ = 3.93, Cohen’s d = 0.29, *p* < 0.001). Overall, most differences were trivial to moderate with no large differences.

Results from the multilevel models are presented in Table 2. A BLL ≥ 2 µg/dL was associated with ~3 points higher score on behavior ratings (3 points are equivalent to ~0.25% of a standard deviation in the T-scores) across all direct effects models. One exception was no association between BLL category and oppositional behavior. We found no evidence of a direct association between any neighborhood factor and child behavior scores. No effect modification of the association between BLL and behavior scores was supported for neighborhood disadvantage or NDVI. There was some evidence for effect modification by distance to nearest greenspace for the association between BLL and planning/organization problems (t = −1.87, *p* = 0.062), oppositional behavior (t = −1.78, *p* = 0.075), cognitive problems (t = −2.01, *p* = 0.045), hyperactivity (t = −2.03, *p* = 0.043), and ADHD (t = −1.92, *p* = 0.055).

**Table 2 toxics-10-00517-t002:** Estimates of direct effects and interaction models across child behavior problem scales and neighborhood effect modifiers.

	Effect Modifier: Neighborhood Disadvantage		Effect Modifier: NDVI		Effect Modifier: Nearest Greenspace (Deciles)	
*Behavior Problem Scale* *Model Variables*	Direct Effects Model	Interaction Term Model	Direct Effects Model	Interaction Term Model	Direct Effects Model	Interaction Term Model
	β (CI)	β (CI)	β (CI)	β (CI)	β (CI)	β (CI)
*BRIEF Planning/Org. Problems*						
Blood Lead ≥ 2 µg/dL	3.5 (0.75, 6.31) *	1.74 (−2.69, 6.17)	3.52 (0.74, 6.30) ^±^	5.37 (−6.14, 16.88)	3.76 (0.96, 6.55) *	8.24 (2.77, 13.71) *
Neighborhood Disadvantage	0.26 (−1.56, 2.08)	−0.32 (−2.47, 1.83)	−	−	−	−
Blood Lead * Neighborhood Dis.	−	1.11 (−1.02, 3.23)	−	−	−	−
NDVI (x100)	−	−	0.03 (−0.16, 0.23)	0.06 (−0.20, 0.32)	−	−
Blood Lead * NDVI	−	−	−	−0.06 (−0.39, 0.28)	−	−
Nearest Greenspace (deciles)	−	−	−	−	0.34 (−0.17, 0.84)	0.75 (0.02, 1.5) *
Blood Lead * Greenspace	−	−	−	−	−	−0.90 (−1.8, 0.05) ^±^
*BRIEF Inhibitory Control Problems*						
Blood Lead ≥ 2 µg/dL	2.83 (0.68, 4.98) *	4.00 (0.48, 7.52) *	2.85 (0.69, 5.01) *	−3.67 (−12.68, 5.32)	2.72 (0.56, 4.88) *	4.47 (0.15, 8.80)
Neighborhood Disadvantage	−0.53 (−2.24, 1.17)	−0.17 (−2.09, 1.75)	−	−	−	−
Blood Lead * Neighborhood Dis.	−	−0.68 (−2.32, 0.96)	−	−	−	−
NDVI (x100)	−	−	−0.03 (−0.19, 0.13)	−0.13 (−0.34, 0.07)	−	−
Blood Lead * NDVI	−	−	−	0.19 (−0.07, 0.46)	−	−
Nearest Greenspace (deciles)	−	−	−	−	−0.19 (−0.58, 0.21)	−0.01 (−0.56, 0.55)
Blood Lead * Greenspace	−	−	−	−	−	−0.33 (−1.05, 0.38)
*CTRS−R:S Oppositional Behavior*						
Blood Lead ≥ 2 µg/dL	2.08 (−0.51, 4.67)	3.72 (−0.32, 7.77)	2.12 (−0.48, 4.71)	3.29 (−13.56, 6.97)	2.00 (−0.59, 4.59)	6.38 (1.27, 11.48) *
Neighborhood Disadvantage	0.48 (−1.55, 2.51)	0.07 (−2.24, 2.39)	−	−	−	−
Blood Lead * Neighborhood Dis.	−	−0.20 (−2.11, 1.71)	−	−	−	−
NDVI (x100)	−	−	−0.04 (−0.23, 0.14)	−0.09 (−0.33, 0.14)	−	−
Blood Lead * NDVI	−	−	−	0.20 (−0.10, 0.49)	−	−
Nearest Greenspace (deciles)	−	−	−	−	−0.23 (−0.69, 0.23)	0.29 (−0.41, 0.98)
Blood Lead * Greenspace	−	−	−	−	−	−0.84 (−1.69, 0.01) ^±^
*CTRS−R:S Cognitive Problems*						
Blood Lead ≥ 2 µg/dL	2.9 (0.03, 5.70) *	3.67 (−0.55, 7.89)	3.11 (0.58, 5.65) *	−1.58 (−12.22, 9.06)	3.29 (0.74, 5.83) *	7.87 (2.82, 12.92) *
Neighborhood Disadvantage	−0.1 (−1.70, 1.46)	0.30 (−1.83, 2.42)	−	−	−	−
Blood Lead * Neighborhood Dis.	−	−0.30 (−2.36, 1.76)	−	−	−	−
NDVI (x100)	−	−	0.08 (−0.09, 0.25)	0.00 (−0.25, 0.25)	−	−
Blood Lead * NDVI	−	−		0.14 (−0.17, 0.44)	−	−
Nearest Greenspace (deciles)	−	−	−	−	0.24 (−0.21, 0.69)	0.78 (0.10, 1.47) *
Blood Lead * Greenspace	−	−	−	−	−	−0.88 (−1.73, −0.03) *
*CTRS−R:S Hyperactivity*						
Blood Lead ≥ 2 µg/dL	2.93 (0.31, 5.55) *	4.3 (−0.51, 9.14)	2.91 (0.29, 5.53) *	−5.17 (−16.40, 6.06)	2.94 (0.31, 5.56) *	7.63 (2.40, 12.87) *
Neighborhood Disadvantage	0.09 (−1.59, 1.78)	0.4 (−2.19, 3.07)	−	−	−	−
Blood Lead * Neighborhood Dis.	−	−0.8 (−2.84, 1.33)	−	−	−	−
NDVI (x100)	−	−	0.02 (−0.16, 0.20)	−0.11 (−0.37, 0.15)	−	−
Blood Lead * NDVI	−	−	−	0.24 (−0.08, 0.56)	−	−
Nearest Greenspace (deciles)	−	−	−	−	0.00 (−0.46, 0.46)	0.57 (−0.13, 1.27)
Blood Lead * Greenspace	−	−	−	−	−	−0.91 (−1.79, −0.03) *
*CTRS−R:S ADHD*						
Blood Lead ≥ 2 µg/dL	3.38 (0.94, 5.81) *	3.72 (−0.29, 7.73) ^±^	3.36 (0.92, 5.80) *	−3.29 (−13.67, 7.08)	3.38 (0.94, 5.82) *	7.51 (2.65, 12.38) *
Neighborhood Disadvantage	−0.06 (−1.59, 1.48)	0.07 (−1.91, 2.06)	−	−	−	−
Blood Lead * Neighborhood Dis.	−	−0.20 (−2.14, 1.74)	−	−	−	−
NDVI (x100)	−	−	0.02 (−0.15, 0.18)	−0.09 (−0.33, 0.14)	−	−
Blood Lead * NDVI	−	−	−	0.20 (−0.10, 0.49)	−	−
Nearest Greenspace (deciles)	−	−	−	−	0.01 (−0.42, 0.44)	0.51 (−0.14, 1.16)
Blood Lead * Greenspace	−	−	−	−	−	−0.80 (−1.61, 0.02) ^±^

**Note.** Models controlled for sex, age, year of enrollment, BMI, hemoglobin, caregiver smoking (yes/no), mother employment status (yes/no), HOME inventory score, maternal education, number of possessions of wealth, and census segment population total. Org.—organizing * *p* < 0.05, ^±^
*p* < 0.10 Figure 1 plots the model estimated means as a function of distance to nearest greenspace, separately for children with BLL ≥ 2 and <2 μg/dL. Overall, children living nearest to greenspace (1st decile = 0.08 km) with BLLs < 2 µg/dL scored 5 to 8-points lower across multiple behavior problem scales compared to children with BLLs ≥ 2 µg/dL. When furthest from greenspace, children were similar on behavior problems regardless of BLL.

Children living nearest to greenspace with BLLs < 2 µg/dL scored 5 to 8 points lower across behavior problem scales compared to children with BLLs ≥2 µg/dL. When furthest from greenspace, children were similar on behavior problems regardless of BLL.

### 3.2. Longitudinal Sample

Sociodemographic characteristics of the longitudinal sample at baseline are presented in Table 3. Participants were balanced on sex (50% male) and of moderately low socioeconomic status: an average 8 years of maternal education, 2.5 possessions, and 56% employed mothers. Tantrums almost daily and frequent conflicts were common (32.9% and 35.5%, respectively). Compared to those excluded from the study, the longitudinal sample participants were from slightly more populous census segments (t_451_ = −3.31, Cohen’s d = 0.57, *p* < 0.01). No other differences in baseline characteristics were noted.

Direct effects and interaction terms for the longitudinal sample are presented in Table 4. Having a BLL ≥2 µg/dL at ~7 years was associated with a ~70% greater likelihood of parent reports of frequent child tantrums at ~8 years. No association was noted between BLL and frequency of conflict. Greater neighborhood disadvantage was associated with 20% lower likelihood of frequent conflicts among caregivers and their children (OR = 0.80 (0.64, 0.97), *p* = 0.028). Greater NDVI was associated with 3% higher likelihood of tantrums almost every day (OR = 1.03 (1.0, 1.06), *p* = 0.036). Distance to a greenspace modified the association between child BLL and the likelihood of tantrums almost every day (*p* = 0.089), and frequent conflicts (*p* = 0.098).

As shown in Figure 2, children living closer to a greenspace were less likely to have a tantrum almost every day and less likely to have frequent parental conflicts if they also had a BLL < 2 µg/dL (tantrums ~20%, conflicts ~30%), compared to children with BLLs ≥2 µg/dL at the same distance (tantrums ~45%, conflicts ~50%). However, children with BLLs ≥2 µg/dL had a similar likelihood of tantrums and parental conflicts to those with BLLs < 2 µg/dL when living farthest from a greenspace (tantrums ~40% [BLL < 2 and ≥2 µg/dL], conflicts ~40% (BLL < 2 µg/dL), 30% (BLL ≥ 2 µg/dL)).

Children living nearest to greenspace with BLLs < 2 µg/dL were ~20% less likely to have tantrums almost every day and frequent parental conflicts compared to children with BLLs ≥ 2 µg/dL. When furthest from greenspace, children were similar on tantrums and parental conflicts regardless of BLL.

## 4. Discussion

Using both cross-sectional and longitudinal data of children with relatively low BLLs (Mean ± SD 2.7 ± 2.11; 52% children ≥ 2 µg/dL), this study examined the extent to which neighborhood factors modify the effects of lead exposure on child behavior. We hypothesized that children living in disadvantaged neighborhoods would have higher problem behavior scores at BLLs ≥ 2 µg/dL. Conversely, we posited that enriching neighborhood factors such as NDVI or access to greenspaces might buffer the relationship between lead and child behavior. Contrary to our hypotheses, we found no evidence that neighborhood disadvantage or NDVI level modifies the association between low-level BLLs, and child behavior measured cross-sectionally or longitudinally. We did, however, find evidence that access to greenspaces may play an important role in the relationship between lead exposure and behavior problems in early school years. Shorter distances to a greenspace were associated with lower behavior scores at ~7 years among children with BLLs < 2 µg/dL, compared to children with BLLs ≥ 2 µg/dL at the same distance. Conversely, at higher distance to greenspace, children with BLLs < 2 µg/dL had similar behavior ratings by teachers as children with BLLs ≥ 2 µg/dL. A similar pattern of differential associations by distance to greenspace was observed for BLLs and caregiver reports of tantrums at ~8 years of age, although these effects were only marginally significant. To summarize, we found that: (i) children with BLL < 2 µg/dL appear to derive additional benefits from living close to enriching neighborhood factors such as greenspaces compared to children BLL ≥ 2 µg/dL, and (ii) children living far from greenspaces had similar teacher and parent-reported behavior scores regardless of their BLLs. While additional research is needed to better understand these results, our findings suggest the need for both child-centered neighborhood design and lead exposure prevention for optimal behavioral health in schoolchildren.

### 4.1. Associations of Lead Exposure and Child Behavior

Children with BLLs ≥ 2 µg/dL had higher teacher ratings of problem behaviors, including planning/organization, inhibitory control, cognitive problems, hyperactivity, and ADHD index. While BLL category was not statistically associated with oppositional behavior, the observed results were in the expected direction. We previously published on the relationship between BLL and child behavior in ~200 children enrolled into SAM between 2009 and 2013 [6]. The current study doubles the sample size, includes a caregiver assessment of child behavior, and moves beyond examining strictly cross-sectional associations. While model-estimated mean T-scores for children with BLLs > 2 µg/dL did not exceed the clinical T-score cut-off, behavior problems persist over time [63,64], subclinical behavior problems in childhood may contribute to other risky behaviors and psychiatric diagnoses later in life [65,66,67], contribute to poor school readiness [68], and are more common in the general population [69]. Thus, our results further support the link between BLLs as low as 2 µg/dL and problematic childhood behaviors.

Caregiver-reported frequent tantrums and conflicts were moderately common at ~8 years of age (33% and 36%, respectively). This is higher than many younger samples, with 10% of 4 year olds having a temper tantrum once per day in the US [70]. Similarly, 57% of Finnish parents reported that their 5 year olds (younger than our sample) no longer had tantrums [71]. The high frequency of reported tantrums and conflicts may be due to sample recruitment. Recruitment focused on neighborhoods with suspected heavy metal exposure. Furthermore, these children come from lower-to-average socioeconomic position where tantrums may be more common [72]. Toxicant mixtures may also play a role. Recently, our novel exposure measurement using silicone wristbands revealed co-exposure to many toxic chemicals including dichlorodiphenyltrichloroethane (DDT), a banned substance in the United States [73]. Future research should investigate how toxicant mixtures influence child behavior in school and at home, including conflictual family relationships that impact overall family functioning [74].

### 4.2. Associations of Neighborhood Factors and Child Behavior

In our cross-sectional analysis, none of the neighborhood factors were associated with child behavior at ~7 years of age. Many previous studies have found positive associations between neighborhood disadvantage and child behavior problems [75,76,77]. Previously, we demonstrated that greater neighborhood disadvantage was associated with higher scores on the CRTS-R:S oppositional behavior scale, BRIEF shifting problems scale, and BRIEF emotional control scale [46]. A complete comparison between the current and previous studies is not possible because BRIEF shifting problems and emotional control scales were not administered in the current study. Furthermore, the effect size in the previous study was relatively small (one standard deviation greater neighborhood disadvantage was associated with 1-point greater behavior ratings).

In the longitudinal study, neighborhood disadvantage was negatively associated with frequent parental conflicts, and NDVI positively associated with tantrums almost every day. These relationships are contrary to many published studies on neighborhood disadvantage [21,22] and NDVI [78,79,80]. To examine the unexpected relationship between neighborhood disadvantage and parental conflicts, we performed a post hoc examination of the conflict question. We examined a related question, which asked caregivers to identify the most common reasons for these conflicts. Caregivers answered “yes” to as many of the following answer options as were applicable: going to sleep, eating, helping in the home, getting up in the morning, doing homework, getting dressed in the morning, friends, or “other”. Caregivers who reported frequent conflicts were equally likely to say “yes” or “no” to all the issues (~50%), except for “friends” (88%). This suggests that peer relationships are the most common reason for conflicts between caregivers and 8-year-old children in our study. Furthermore, the negative association between neighborhood disadvantage and frequent conflicts could be related to differences in parenting practices among families of low socioeconomic position. While parents of high socioeconomic position often encourage independent thinking and questioning, obedience can be adaptive for children from contexts where levels of violence and crime are high [81]. Thus, parental conflicts may be less frequently reported in families from more disadvantaged settings. Complicating this further, the cultural dynamics of neighborhood, peer and parental relationships in Montevideo may be quite different from a US context. Future research should consider cultural context when evaluating the relationship between neighborhood disadvantage and parent-child conflict. As the relationship between NDVI and tantrums almost every day was small, it may be a spurious finding. Once again, however, cultural differences in how tantrums are perceived should encourage caution when interpreting these results.

### 4.3. Effect Modification by Neighborhood Factors

It is theorized that children from disadvantaged neighborhoods may have a heightened stress response to lead exposure, thereby exacerbating behavior problems [82]. In this sample of 7 and 8 year olds with low-level lead exposure, we did not find support for effect modification by neighborhood disadvantage. To the best of our knowledge, this is the first study to test this theory in a sample of lead-exposed children. While we did not find effect modification by our measure of neighborhood disadvantage, other neighborhood-related factors may play a role. For example, exposure to community violence may be an important source of stress as violent crime in Uruguay has increased in recent years [83]. For example, a recent mixed-methods study revealed that Uruguayan adolescents from underprivileged Montevideo neighborhoods report frequent exposure to community violence [84]. We did not include crime in the neighborhood disadvantage factor; future research may benefit from the inclusion of factors more directly related to stress such as community violence and crime.

We did find effect modification between child BLL and all behavior scores by distance to nearest greenspace. First, children with BLLs < 2 µg/dL demonstrated lower behavior problems when living near greenspace compared to children with BLLs ≥ 2 µg/dL at the same distance. Behavior scores of children with BLLs < 2 µg/dL and BLLs ≥ 2 µg/dL were similar when farthest from a greenspace. The effect modification by distance to greenspace was robust, demonstrated across scales, via cross-sectionally measured teacher reports, and a year later using caregiver reports. While these findings suggest that better access to greenspaces may not buffer the effects of BLL ≥ 2 µg/dL on behavior, children with BLLs < 2 µg/dL may uniquely benefit from access to greenspace.

It is important to consider how these effect modifiers were operationalized. Neighborhood disadvantage was assigned at the census segment level, which may be much larger than the child’s perceived neighborhood area. Our prior work suggested that caregiver-reported qualitative assessments of neighborhood boundaries were much smaller than an average census segment in Montevideo [46]. Furthermore, the amount of time a child spends in a geographic area may affect exposure duration and intensity for neighborhood factors.

Distance to nearest greenspace but not NDVI was an effect modifier of BLL on child behavior. These two measures were only minimally correlated in the cross-sectional sample (Spearman’s Rho = 0.15). We used satellite imagery and a 150 m buffer to measure NDVI. Both larger and smaller buffer sizes have been used when assessing relationships with child behavior [78,79]. We did not test varying buffer distances to prevent inflating type 1 error rates. Air pollution is diminished in areas of high greenness, which may decrease exposure to airborne lead [85].

Utilizing alternative NDVI buffers, measuring participants’ perceptions of greenspace [86], and use of greenspace in their neighborhoods [87] may help refine exposure to greenspace. For example, GPS monitoring and wearable cameras may help assess time and activities in differing greenspaces [88,89]. By contrast, greenspace was measured using photogrammetry and included landscaped areas, gardens, parks, and other green spaces. The surrounding green foliage around a child’s home measured via NDVI may not reflect access to larger greenspaces, which may be better for physical activity and socialization. Improving measurements of greenspace exposure may help differentiate the influence of greenspace activities vs. green foliage.

While public health efforts have correctly focused on preventing lead exposure, lack of access to greenspaces may have similar impacts on child behavior. Neighborhood level interventions to reduce behavior issues should consider multifaceted approaches that include investment in urban design as well as lead remediation. Future research might consider what kinds of greenspaces are best for child development.

### 4.4. Strengths and Limitations

To the best of our knowledge, this is the first study to examine potential effect modification of lead exposure by multiple neighborhood factors. Several behavior measures were administered across two types of informants, with teachers reporting on specific behavior problems (oppositionality, hyperactivity, ADHD) and on behaviors relevant to school success (cognitive problems, planning/organization, and cognitive shifting). Caregivers also reported on parent–child relationships. We examined both detrimental (neighborhood disadvantage) and enriching (NDVI, distance to greenspace) neighborhood factors when evaluating potential effect modification. Furthermore, lead exposure was not associated with neighborhood disadvantage in our sample, allowing for an opportunity to examine their unique effects that are often difficult to tease apart in in US-based samples where neighborhood disadvantage and childhood lead exposure overlap.

We do note some limitations. Our sample had low levels of lead exposure. However, mean BLLs were ~3 times higher than current levels in US children. While broad generalizations of our findings may not be appropriate for samples with significantly lower BLLs, levels of lead exposure in our sample were appropriate to assess effect modification. Future studies may replicate our findings among samples with even lower levels of lead exposure. We tested a limited number of neighborhood factors that potentially act as effect modifiers. For example, exposure to community violence, unmeasured in our sample, could be further explored. Novel measurement approaches that consider duration and frequency of greenspace use should also be incorporated. Our selected cross-sectional analytical sample had a higher percentage of children with BLLs ≥ 2 µg/dL, coming from neighborhoods with greater disadvantage and shorter distance to greenspaces compared to non-selected children. These selection effects were generally small, but it is possible they muted the associations between BLL, neighborhood disadvantage and greenspace access on behavior problems. A larger sample with wider ranges of blood lead, neighborhood disadvantage and greenspace access may have yielded larger effect sizes. In the same vein, because we tested effect modification, we may have been underpowered to detect smaller effects. A larger sample may also allow for testing three-way interactions between BLL, neighborhood disadvantage and greenspace access. Furthermore, as BLLs decline globally, efforts should be focused on investigating these relationships in the context of low or very low lead exposure, which may also require larger samples. Finally, we note that BLLs were measured at one time point and may not reflect lifetime exposure or exposure during earlier critical windows of development.

In summary, even at low levels of lead exposure, children with BLLs ≥ 2 µg/dL had higher behavior problems. Children with BLLs < 2 µg/dL at the farthest distance from greenspace had similar behavior ratings as children with BLLs ≥ 2 µg/dL at the same distance. Conversely, children with BLLs < 2 µg/dL had much lower behavior problem scores compared to children with BLLs ≥ 2 µg/dL when nearest to greenspace. While greenspace access did not appear to buffer the detrimental effects of BLLs ≥ 2 µg/dL at ~7 years of age, children with BLLs < 2 µg/dL seem to derive additional benefits of living closer to green spaces. Additional research is needed to clarify critical modifying factors that could form the basis for multifactorial neighborhood interventions to prevent behavior issues in school age children.

## Figures and Tables

**Figure 1 toxics-10-00517-f001:**
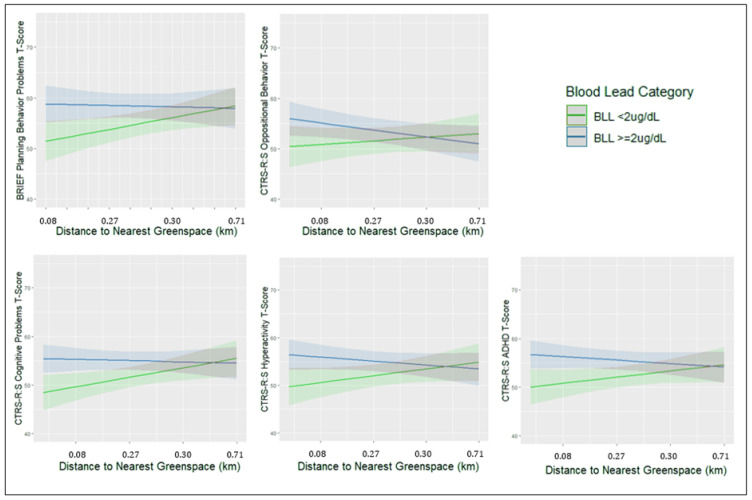
Cross-sectional effect modification of BLL by neighborhood factors on child behavior ratings.

**Figure 2 toxics-10-00517-f002:**
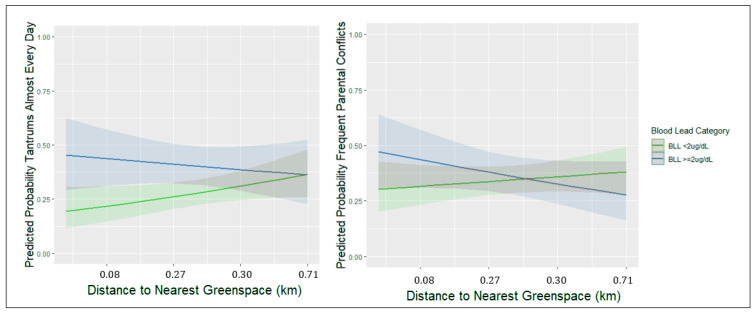
Longitudinal effect modification of BLL by caregiver reported tantrums and parental conflicts.

**Table 1 toxics-10-00517-t001:** Characteristics of cross-sectional sample compared with non-selected participants.

Variables	BRIEF Analytical Sample		BRIEF Non-Selected Participants			CTRS-R:S Analytical Sample		CTRS-R:S Non-Selected Participants		
*Covariates*	Statistic	*n*	Statistic	*n*	Standard Difference		*n*		*n*	Standard Difference
% Male	55.1%	448	49.0%	237	*phi* = 0.06	53.9%	362	52.0%	323	*phi* = 0.02
Age in Months, Mean (SD)	81.7 (6.08) **	448	83.2 (6.02)	236	d = 0.25	81.4 (6.18) ***	362	83.1 (5.88)	322	d = 0.28
Maternal Education in Years, Mean (SD)	8.6 (2.52)	442	8.2 (2.40)	228	d = 0.16	8.7 (2.49) **	356	8.2 (2.47)	314	d = 0.20
Number of Possessions of Wealth, Mean (SD)	2.9 (1.34) *	427	2.7 (1.26)	223	d = 0.15	3.0 (1.36) *	341	2.7 (1.27)	309	d = 0.23
HOME Inventory Score, Mean (SD)	41.7 (9.31) **	379	38.9 (11.15)	201	d = 0.27	42.1 (8.88) ***	298	39.3 (11.02)	282	d = 0.28
Mother Employed (% Yes)	62.7%	394	57.6%	203	*phi* = 0.05	64.6%	316	56.9%	281	*phi* = 0.08
Either Caregiver Smokes (% Yes)	60.2%	425	63.9%	227	*phi* = 0.04	58.1%	339	65.2%	313	*phi* = 0.07
Hemoglobin (g/dL), Mean (SD)	13.2 (0.95)	447	13.2 (0.88)	234	d = 0.00	13.2 (1.00)	360	13.2 (0.86)	321	d = 0.0
Body Mass Index, Mean (SD)	16.8 (2.66)	444	16.8 (2.60)	234	d = 0.00	16.7 (2.54)	360	16.9 (2.75)	318	d = 0.08
Population in Census Segment, Mean (SD)	2240.5 (1412.2)	448	2097.4 (1412.2)	237	d = 0.10	2282.2 (1132.2) *	362	2088.7 (1339.6)	323	d = 0.16
*Primary Exposure*										
% Blood Lead ≥ 2 µg/dL	52.0%	448	44.7%	237	*phi* = 0.07	60.2%***	362	37.5%	323	*phi* = 0.23
*Effect Modifiers*										
Neighborhood Dis., Mean (SD)	1.66 (1.19) ***	448	1.10 (0.93)	237	d = 0.52	1.64 (1.19) ***	362	1.28 (1.04)	323	d = 0.32
Nearest Greenspace (km), Mean (SD)	0.22 (0.19) ***	448	0.29 (0.22)	237	d = 0.34	0.21 (0.20) ***	362	0.27 (0.21)	323	d = 0.29
NDVI 150 Meter Buffer, Mean (SD)	0.34 (0.08)	448	0.33 (0.08)	237	d = 0.0	0.34 (0.08) *	362	0.32 (0.08)	323	d = 0.25
*Outcome*										
BREIF Inhibitory Control Problems, Mean (SD)	56.8 (14.88)	448	-		-	-		-		-
BRIEF Planning/Org. Problems, Mean (SD)	52.7 (12.04)	448	-		-	-		-		-
CTRS-R:S Oppositional Behavior, Mean (SD)	-		-		-	53.4 (13.10)		-		-
CTRS-R:S Cognitive Problems, Mean (SD)	-		-		-	54.0 (12.81)		-		-
CTRS-R:S Hyperactivity, Mean (SD)	-		-		-	53.8 (12.28)		-		-
CTRS-R:S ADHD Index, Mean (SD)	-		-		-	54.1 (11.79)		-		-

**Note**. Analytical samples pulled from final *n* of 685 cross-sectional participants that fulfilled inclusion criterion. Row *n*’s may not sum to 685 due to missing data. Org.—organizing; Dis—disadvantage; µg/dL—micrograms per deciliter; SD—standard deviation; d = Cohen’s d; *phi* = *phi* coefficient; *n* = sample size * *p* < 0.05; ** *p* < 0.01; *** *p* < 0.001.

**Table 3 toxics-10-00517-t003:** Cohort characteristics for caregiver behavior report follow-up. Comparisons with non-selected participants.

Variables	Caregiver Child Behavior Report Analytical Sample		Caregiver Child Behavior Report Non-Selected Participants		
* **Covariates** *	**Statistic**	*n*	**Statistic**	* **n** *	**Standard Difference**
% Male	49.5%	380	55.8%	52	*phi* = 0.04
Age in Months, Mean (SD)	82.94 (5.74)	380	84.39 (6.02)	51	d = 0.25
Maternal Education in Years, Mean (SD)	8.07 (2.32)	375	7.78 (2.40)	49	d = 0.12
Number of Possessions of Wealth, Mean (SD)	2.53 (1.30)	377	2.76 (1.05)	42	d = 0.19
HOME Inventory Score, Mean (SD)	37.79 (10.29)	314	39.70 (1.45)	47	d = 0.26
Mother Employed (% Yes)	55.5%	312	54.4%	46	*phi* = 0.01
Either Caregiver Smokes (% Yes)	66.3%	380	55.8%	43	*phi* = 0.07
Hemoglobin (g/dL), Mean (SD)	13.30 (0.79)	379	13.14 (0.90)	50	d = 0.24
Body Mass Index, Mean (SD)	16.81 (2.72)	376	16.78 (2.25)	50	d = 0.01
Population in Census Segment, Mean (SD)	2369.30 (1433.55) **	380	1694.98 (871.98)	52	d = 0.57
*Primary Exposure*					
% Blood Lead ≥ 2 µg/dL	34.0%	380	23.1%	52	*phi* = 0.08
*Effect Modifiers*					
Neighborhood Dis., Mean (SD)	1.20 (1.00)	380	1.49 (1.10)	52	d = 0.28
Nearest Greenspace (km), Mean (SD)	0.26 (0.20)	380	0.29 (0.26)	52	d = 0.13
NDVI 150 Meter Buffer, Mean (SD)	0.33 (0.08)	380	0.34 (0.09)	52	d = 0.12
*Outcome*					
% Tantrums Almost Every Day	32.9%	380	-	-	-
% Parental Conflicts Frequently	35.5%	380	-	-	-

**Note**. Analytical samples pulled from enrolled cohort participants (*n* = 432) that fulfilled inclusion criterion. Row *n*’s may not sum to 432 due to missing data. Dis.—disadvantage; µg/dL—micrograms per deciliter; SD—standard deviation; d = Cohen’s d; *phi* = *phi* coefficient; ** *p* < 0.01.

**Table 4 toxics-10-00517-t004:** Estimates of direct effects and interaction models across caregiver report follow-up and neighborhood effect modifiers.

	Effect Modifier: Neighborhood Disadvantage		Effect Modifier: NDVI		Effect Modifier: Nearest Greenspace (Deciles)	
*Caregiver Report Follow-Up* *Model Variables*	Direct Effects Model	Interaction Term Model	Direct Effects Model	Interaction Term Model	Direct Effects Model	Interaction Term Model
	OR (CI)	OR (CI)	OR (CI)	OR (CI)	OR (CI)	OR (CI)
*Tantrums Almost Every Day*						
Blood Lead ≥ 2 µg/dL	1.7 (1.1, 2.8) ^±^	1.8 (0.85, 4.01)	1.8 (1.1, 2.8) *	0.4 (0.06, 3.41)	1.7 (1.10, 2.80) *	3.96 (1.37, 11.45) *
Neighborhood Disadvantage	1.0 (0.79, 1.21)	1.0 (0.76, 1.31)	-	-	-	-
Blood Lead * Neighborhood Dis.	-	1.0 (0.63, 1.44)	-	-	-	-
NDVI (x100)	-	-	1.03 (1.0, 1.06) *	1.0 (0.98, 1.05)	-	-
Blood Lead * NDVI	-	-	-	1.0 (0.98, 1.11)	-	-
Nearest Greenspace (deciles)	-	-	-	-	1.04 (0.96, 1.13)	1.1 (1.01, 1.23) *
Blood Lead * Greenspace	-	-	-	-	-	0.87 (0.74, 1.02) ^±^
*Parental Conflicts Frequently*						
Blood Lead ≥ 2 µg/dL	1.0 (0.88, 1.20)	0.9 (0.44, 1.94)	1.1 (0.69, 1.75)	1.1 (0.17, 7.60)	1.1 (0.68, 1.72)	2.33 (0.84, 6.43)
Neighborhood Disadvantage	0.8 (0.64, 0.98) *	0.8 (0.59, 0.99) *	-	-	-	-
Blood Lead * Neighborhood Dis.	-	1.1 (0.73, 1.68)	-	-	-	-
NDVI (x100)	-	-	1.02 (0.99, 1.05)	1.02 (0.99, 1.06)	-	-
Blood Lead * NDVI	-	-	-	1.0 (0.94, 1.06)	-	-
Nearest Greenspace (deciles)	-	-	-	-	1.0 (0.92, 1.10)	1.0 (0.95, 1.14)
Blood Lead * Greenspace	-	-	-	-	-	0.9 (0.75, 1.03) ^±^

**Note**. Models controlled for sex, age, BMI, hemoglobin, caregiver smoking (yes/no), mother employment status (yes/no), HOME inventory score, maternal education, number of possessions of wealth, and census segment population total. * *p* < 0.05, ^±^
*p* < 0.10.

## Data Availability

Data available upon request.

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
