# Peer review of "Do Neighborhood Factors Modify the Effects of Lead Exposure on Child Behavior?"

_toxics, 2022, doi:10.3390/toxics10090517_

Round 1

Reviewer 1 Report

In this manuscript by Frndak et al., the authors address the question whether neighborhood factors modify the effects of lead exposure on child behavior, taking samples of BLLs from 7-year old children in Uruguay. This is a well-written and well-developed manuscript, the method was well explained and the results are clear. I recommend it for publication in Toxics.

Author Response

Reviewer 1

In this manuscript by Frndak et al., the authors address the question whether neighborhood factors modify the effects of lead exposure on child behavior, taking samples of BLLs from 7-year old children in Uruguay. This is a well-written and well-developed manuscript, the method was well explained and the results are clear. I recommend it for publication in Toxics.

Thank you for this kind review of our work.

Reviewer 2 Report

This is an interesting and well conducted study focused on neighborhood factors as effect modifiers of the lead-behavior relationship. The manuscript is already very well written and described in details in all sections. The only limitation that should be discussed is that lead exposure is based on one BLL measurement on samples collected at one point. This may not reflect the overall exposure in the previous years of age, and does not provide information about more vulnerable previous exposure windows.

- on line 38: I would specify 'persistent negative associations'

- on line 39: 'the Centers for Disease Control and Prevention lowered in 2021'

Author Response

Reviewer 2

This is an interesting and well conducted study focused on neighborhood factors as effect modifiers of the lead-behavior relationship. The manuscript is already very well written and described in details in all sections. The only limitation that should be discussed is that lead exposure is based on one BLL measurement on samples collected at one point. This may not reflect the overall exposure in the previous years of age and does not provide information about more vulnerable previous exposure windows.

Thank you for your kind review. We have added this suggestion about BLL measurement as part of the discussion.

- on line 38: I would specify 'persistent negative associations'

Thank you. This has been added.

- on line 39: 'the Centers for Disease Control and Prevention lowered in 2021'

Thank you. This has been added.

Reviewer 3 Report

Comment 1:Demonstrate in detail the association between BLLs and child behavior.

Comment 2: Describe random forest imputation and normalized difference vegetation index in detail.

Comment 3: Abstract need to reflect the importance of study, methods used, outlook presented and results.

Comment 4: Captions for figures need to be more detailed and consist of more details. Correct this.

Comment 5: please discuss the implications of your research in depth.

Comment 6: English language should be corrected by a professional lector.

Comment 7: Describe neighborhood disadvantage index in detail.

Author Response

We thank the reviewer for the comments below. We made the best possible attempt to address them but as they do not provide much explanation of what specifically needs to be changed, we may have missed the mark. Additionally, given the length of the paper, we were trying to be concise in our revisions. We do hope, however, that the changes we made are satisfactory.

Comment 1:Demonstrate in detail the association between BLLs and child behavior.

We have added information describing neurologic mechanisms of lead exposure that might account for the association between BLLs and child behavior. .

Comment 2: Describe random forest imputation and normalized difference vegetation index in detail.

We added sentences describing random forest in more detail and now provide the NDVI formula.

Comment 3: Abstract need to reflect the importance of study, methods used, outlook presented and results.

We made a few edits to the abstract in hopes of addressing this concern.

Comment 4: Captions for figures need to be more detailed and consist of more details. Correct this.

We have added captions to the figures.

Comment 5: please discuss the implications of your research in depth.

Thank you for this suggestion. We discuss the implication of our work in lines 537-540 and 651-655. As this request is not very specific, we are open to further guidance from the reviewer beyond what we have already provided.

Comment 6: English language should be corrected by a professional lector.

Thank you. The manuscript has been reviewed as suggested and we believe all issues of misspelling or missing words have been resolved.

Comment 7: Describe neighborhood disadvantage index in detail.

Thank you for this suggestion. We have cited the original paper that describes the creation of this factor in detail.

Reviewer 4 Report

The authors describe how more than half of the original cohorts were not included in the study, in what appears to be a transparent way.

A point of improvement is that most of the references are dated to very dated. I would recommend using newer studies as references, as much as possible.

Author Response

Reviewer 4

The authors describe how more than half of the original cohorts were not included in the study, in what appears to be a transparent way.

A point of improvement is that most of the references are dated to very dated. I would recommend using newer studies as references, as much as possible.

Thank you. We have included updated references.